# Theoretical Study on Symmetry-Broken Plasmonic Optical Tweezers for Heterogeneous Noble-Metal-Based Nano-Bowtie Antennas

**DOI:** 10.3390/nano11030759

**Published:** 2021-03-17

**Authors:** Guangqing Du, Yu Lu, Dayantha Lankanath, Xun Hou, Feng Chen

**Affiliations:** State Key Laboratory for Manufacturing System Engineering and Shaanxi Key Laboratory of Photonics Technology for Information, School of Electronic Science and Engineering, Xi’an Jiaotong University, Xi’an 710049, China; guangqingdu@mail.xjtu.edu.cn (G.D.); zjkly19900714@126.com (Y.L.); dyanthalankanath1@gmail.com (D.L.); houxun@mail.xjtu.edu.cn (X.H.)

**Keywords:** optical tweezers, bowtie dimer, heterogeneous, trapping potential

## Abstract

Plasmonic optical tweezers with a symmetry-tunable potential well were investigated based on a heterogeneous model of nano-bowtie antennas made of different noble substances. The typical noble metals Au and Ag are considered as plasmonic supporters for excitation of hybrid plasmonic modes in bowtie dimers. It is proposed that the plasmonic optical trapping force around a quantum dot exhibits symmetry-broken characteristics and becomes increasingly asymmetrical with increasing applied laser electric field. Here, it is explained by the dominant plasmon hybridization of the heterogeneous Au–Ag dimer, in which the plasmon excitations can be inconsistently modified by tuning the applied laser electric field. In the spectrum regime, the wavelength-dependent plasmonic trapping potential exhibits a two-peak structure for the heterogeneous Au–Ag bowtie dimer compared to a single-peak trapping potential of the Au–Au bowtie dimer. In addition, we comprehensively investigated the influence of structural parameter variables on the plasmonic potential well generated from the heterogeneous noble nano-bowtie antenna with respect to the bowtie edge length, edge/tip rounding, bowtie gap, and nanosphere size. This work could be helpful in improving our understanding of wavelength and laser field tunable asymmetric nano-tweezers for flexible and non-uniform nano-trapping applications of particle-sorting, plasmon coloring, SERS imaging, and quantum dot lighting.

## 1. Introduction

In recent years, nano-scale optical tweezers, which could allow the ultra-accurate trapping of single nano-objects, have attracted a great deal of attention because of their potential ability to precisely manipulate nanoparticles for scientific and practical purposes [1,2,3,4,5]. Among the strategies for nano-scale trapping, plasmonic excitation in nano-structures to squeeze light into nanoscale offers potential to engineer nano-tweezers for specific applications such as molecule manipulation, particle sorting, quantum dot lighting, etc. Physically, the localized electric field can be excited by electron-rich materials with respect to specific geometries matching to laser excitations [6,7,8,9,10]. The plasmonic force is generated from plasmon resonant nanostructures due to the enhanced electric field at geometries such as nano-trench, nano-dot, bowtie dimer, and ring-slit. Amongst these, the bowtie dimer is the most promising plasmon geometry for high-precision optical nano-tweezer applications, due to its wide excitation spectrum and giant electric field enhancements [11,12,13,14]. Previous investigations into plasmonic trapping have mainly focused on homogeneous bowtie dimers for stable and high-precision trapping purposes [15,16,17,18]. The optical forces generated from a gold bowtie antenna were numerically investigated by Cetin [19], who found that much larger optical forces could be produced when the incident light polarization was along the bowtie gap. Lin et al. theoretically examined the plasmonic bowtie notch, achieving excellent trapping capability with ultralow threshold excitation intensity [20]. Roxworthy et al. experimentally explored plasmonic trapping properties using Au bowtie nanoantenna arrays, finding that the measured optical trapping efficiencies were up to 20× the efficiencies of conventional high-NA (Numerical Aperture )optical traps [21]. This potentially benefits applications requiring the high-precision trapping of quantum-scale nanoparticles. However, for applications such as sorting non-uniform nanoparticles for display coloring and large-molecule imaging, it is likely that the plasmonics can be spatially tuned to facilitate the flexible manipulation of nanoparticles. Recently, it has been reported that the optical properties of heterodimers in the spectra can be controlled from visible to near-ultraviolet [22,23]. Our previous studies have shown that heterogeneous bowtie dimer plasmonics can give flexible solutions by asymmetrically arranging the localized plasmonic electric field [24]. For the optical trapping processes, it would be even more complex for heterogeneous bowtie dimers due to the pluralistic plasmon excitations required to support nanoscale optical forces at such heterogeneous antenna. In particular, as the trapping particle size decreases to sizes comparable to large molecular and quantum dots, the stable trapping would become even weaker in the heterogeneous bowtie dimer because of dominant Brownian motion for sub-wavelength particles. While heterogeneous plasmonics are being considered for the trapping of very small nanoparticles, little is currently known regarding the non-uniform tunability of noble-metal-based nano-bowtie antennas for stable optical trapping. 

In this paper, we theoretically investigated symmetry-tunable plasmonic trapping properties of nano-bowtie-antenna-based optical tweezers via numerous simulations using the finite element method (FEM). Comparative studies on the plasmonic trapping properties of homo- and hetero-plasmonic bowtie dimers were carried out with respect to the crucial laser excitation parameters of incident wavelength, laser energy, and incident angle. The symmetry-tunable plasmonic potential well properties of the heterogeneous nano-bowtie system were explored in detail. The plasmon hybridization interactions are proposed to explain the symmetry tunability of the plasmonic trapping phenomena. The results are potentially important for applications in the field of non-uniform nanoparticles sorting, including large-molecule imaging and quantum coloring.

## 2. Modeling and Methods

A 2-D plasmonic noble metal nano-bowtie geometry optical tweezers model for the comprehensive prediction of plasmonic trapping properties is proposed, in which the nano-trapping geometry is made of homogeneous or heterogeneous noble bowtie dimers. Physically, as the plane-wave light with vector ”k” is incident on the bowtie dimer, the electron systems of the noble antenna are initially excited in the bowtie system on a femtosecond timescale. The optical tweezers system then becomes electronically resonant as the laser wavelength matches to the surface plasmon frequency determined by the specific designed geometry. We carefully consider the plasmonic optical forces exerted on the Au sphere by building a self-consistent Helmholtz equation model integrated with optical force calculations from three types of contributions: gradient forces, radiation pressure force, and polarization force [25]. The gradient distribution of the localized electric field plays a crucial role in generating the plasmonic trapping force within the resonant bowtie dimer. Furthermore, the plasmon polarizability is a key factor that characterizes the optical response due to the plasmonic interaction between the optical field and nanostructure that determines the strength of the plasmonic optical trapping properties. Plasmonic coupling causes localized electric field enhancement, which augments the trapping force. Rayleigh approximation was taken into account throughout the investigation for all trapped particles which are smaller than the wavelength of the incident trapping laser. The plasmonically enhanced local electric field is confined within the near-field of the trap, generating a considerably strong field gradients and large local field intensities, in return augmenting optical forces acting on the trapped particle to create a strong stable optical trap. Optical forces are calculated based on the dipole approximation, which can be determined by considering a nanosphere as follows [26]:(1)F=14ε0Reα0∇E2+nσ2cE×H*+σ2Reiε0k0E·∇E*
where $\varepsilon_{0}$ is vacuum dielectric permittivity, *α*_0_ is the polarizability of a point-like particle, *E* is the incident electric field, n is the refractive index of the surrounding medium, σ is the total cross section of the particle, c is the speed of light, *H* is the incident magnetic field, and k_0_ is the laser wave vector in free space. Here, *E** and *H** means the conjugate of *E* and *H*, respectively. Re means the real part of a function. 

The first term is the gradient force that provides the spatial confinement in the optical tweezers, which dominates the second and third terms. The second term is identified as the radiation pressure force proportional to the Poynting vector, and the third term is a force arising from the presence of spatial polarization gradients. The combination of the second and the third terms gives the total scattering force. In fact, the electric field force can be much larger than the magnetic force in the visible spectrum. Especially, the plasmonic trapping can be supported in the form of significantly enhanced electric fields comparted to the magnetic field. As a result, the electric field gradient forces from both the spatial and polarization contributions play a crucial role in generating the plasmonic trapping force generated from the resonant noble bowtie dimer.

The trapping potential resulting from the optical forces is a key factor that determines the stability of the optical trap, and it can be obtained by [27]
(2)$$U\left(r_{0}\right)=\int_{\infty }^{r_{0}}F\left(r\right)\cdot dr$$

Numerically, we first built the geometry of the bowtie dimer surrounded by ambient medium using the Comsol Multiphysics built-in geometry drawing function. Then, the bowtie geometry and ambient medium were all divided into many small meshes. The Helmholtz equation was then discretized at every mesh point to form a large sparse matrix on the defined geometry. A perfect matching layer (PML) was set outside of geometrical bowtie dimer. The scattered light from the bowtie was completely absorbed through the PML in the far field. The boundary condition at the interface between the metallic bowtie dimer and ambient medium was treated as continuous. We obtained the numerical solutions of the Helmholtz equation and optical force modeling by solving the built matrix using the powerful FEM solver of Comsol Multiphysics. In fact, we built the dimer systems of Au–Ag, Au–Au, and Ag–Ag based on both the material parameters and geometric design. The material parameters in the current simulations relate to the permittivity of Au and Ag. The interaction between the particle trap system and the trapping laser leads to the generation of optical forces on the particle. In this investigation we focused on local electric field behavior for the trapping force characteristics, symmetry-tunability of trapping potential, and optical tweezers characteristics with varying laser parameter. 

## 3. Results and Discussion

2-D images of plasmonic electric field enhancement and the reinforced optical force for the homogenous Au–Au bowtie dimer with respect to different laser wavelengths are shown in Figure 1. The laser is linearly polarized in the *x* direction and propagates along *y* axis with an incident angle of 0°. The bowtie edge length is set to 20 nm and an Au sphere with 5 nm trapping radius is selected for the purpose of plasmonic trapping applications with small quantum dots in the fields of plasmon coloring, quantum dot lighting, etc. In Figure 1 it can be seen that the plasmonic electric field is dominantly concentrated between the bowtie gap for laser wavelengths of 545 nm (a), 560 nm (b), and 600 nm (c). The maximal plasmonic electric-field image emerged for the excitation wavelength of 560 nm in the current simulation. However, as the excitation wavelength shifts from the central 560 nm, the electric field enhancement is attenuated due to the mismatching plasmon resonance for the bowtie dimer at wavelengths of 545 and 600 nm. The reinforced trapping forces exerted on the Au nanosphere at the central excitation wavelength of 560 nm are presented in normalized form in Figure 1d. Interestingly, it is observed that the dipolar force dominantly occurs in the *x* direction (Figure 1e), while the quadrupole force obviously appears in the *y* component (Figure 1f). As a result, the 2-D normalized trapping force exhibits almost fully symmetrical reinforcement around the trapping Au nanosphere, as in Figure 1d. These results can be helpful for helping to understand the origin of optical trapping behavior tuning via laser wavelength modulation, facilitating the nanoscale plasmonic trapping applications of large molecules and quantum dots.

Figure 2 shows the 2-D images of the plasmonic trapping potential well of the homogeneous Au–Au bowtie dimer with respect to laser excitation wavelengths and applied laser electric fields. It shows that the potential well is evidently strengthened at the excitation wavelength of 560 nm (Figure 2b). Nevertheless, it leads to obvious weakening of trapping potential well at the red-shifted wavelength of 600 nm (Figure 2c). Here, the applied laser electric field amplitude is fixed as *E* = 4 × 10^6^ V/m. 

The calculated trapping potential reaches 14.2 × 10^−21^ J at the working wavelength of 560 nm, which is three times larger than the Brownian motion energy taken as K_B_T. In order to achieve higher stability of plasmonic trapping, an amplified laser field is usually required in order to increase the light-force conversion efficiency to overcome Brownian motion. We observed that the plasmonic trapping potential exhibits dramatic enhancement with increasing laser electric field from 2 × 10^6^ V/m to 4 × 10^6^ V/m and 6 × 10^6^ V/m, as in Figure 2d–f, respectively. When the laser electric field is at 2 × 10^6^ V/m, the trapping potential well becomes insignificant and disappears in the bowtie gap (Figure 2d). The enhanced trapping potential reaches 57 × 10^−21^ J at the laser electric field of 6 × 10^6^ V/m (Figure 2f), which is 12 times larger than K_B_T. This indicates that the trapping potential can work well to overcome the Brownian motion of the Au sphere for stable trapping of the Au sphere target via amplifying the laser field. These results could be fundamentally important for a good understanding of the optimization of laser excitation parameters for obtaining stable plasmonic trapping in bowtie antennas for specific quantum-dot-scale optical trapping applications. In recent years, experimental studies have shown that more abundant optical and electronic properties can be explored for nanoparticle-decorated triangles for potential applications in SERS and hot carrier devices [28,29]. Experimental advancements would significantly accelerate the development of plasmonic trapping applications in a wide range of experimental and industrial fields.

The plasmonic trapping potential well distributions in the *x* direction along the gap of the homogenous and heterogeneous dimer geometries for different laser electric fields are shown in Figure 3. It can be seen from (a) that the plasmonic trapping potential distribution exhibits symmetry across the gap of the homogenous Ag–Ag bowtie at the excitation wavelength of 433 nm. The 433 nm excitation wavelength is selected to improve the trapping robustness compared to the 410 nm resonance wavelength, but simultaneously brings about the sacrifice of trapping potential value. The maximal trapping potential of 27 × 10^−21^ J appears in close proximity to the inner bowtie tips at the excitation wavelength of 433 nm, which is five times larger than K_B_T, at a laser field of 3 × 10^5^ V/m. The smaller laser field amplitudes of 2 × 10^5^ V/m and 1 × 10^5^ V/m all lead to lower trapping well potential for the plasmonic bowtie antennas. However, we find that the trapping potential symmetry has nothing to do with modifying the applied laser field. We can see from Figure 3b that the potential symmetry is substantially broken for the heterogeneous Au–Ag bowtie dimer at the excitation wavelength of 433 nm. More interestingly, we can see that the trapping potential becomes more asymmetrical at the larger laser electric field of 18 × 10^5^ V/m. It can be explained by dominant plasmon hybridization originating from the heterogeneous noble dimers. The hybrid plasmon causes an asymmetric electric field gradient, leading to asymmetrical trapping potential across the bowtie’s gap. To explore the symmetry-tunable properties for robust plasmonic trapping, we further carried out comparable investigations on the potential well for Au–Au, Au–Ag, and Ag–Ag dimers. It can be seen from Figure 3c that a strong symmetrical potential well is observed for the Au–Au dimer at a resonance wavelength of 560 nm. A 2-D image of the symmetry-broken trapping potential distribution for the heterogeneous Au–Ag bowtie dimer is shown as an inset in Figure 3c. The potential well is dominantly concentrated at the region close to the Au tip, indicating that the Au plasmon resonance is playing a role at a wavelength of 560 nm. However, the weakest plasmonic trapping potential appears at the Ag–Ag bowtie dimer due to the off-resonance laser excitation at the excitation wavelength of 560 nm, which is typically deviated from the plasmon frequency of Ag. Note that the Au–Ag bowtie dimer always exhibits a moderate potential well with symmetry-broken distribution. This work could be helpful in developing our understanding of symmetry-tunable plasmonic nano-tweezers for stable and robust plasmonic nano-trapping applications. 

The electric-field enhancement spectra and the wavelength-dependent plasmonic trapping potential tunability for homogeneous Au–Au and heterogeneous Au–Ag bowtie dimers are shown in Figure 4. We can see from (a) that a dual-peak spectrum centered at wavelengths of 415 nm and 549 nm for the electric field enhancement appears for the heterogeneous Au–Ag bowtie dimer. More interestingly, the Fano-resonance profile is definitely observed for the heterogeneous Au–Ag bowtie dimer. It is attributed to the plasmon hybridization for supporting interference between the Au–Ag bowtie plasmons. However, for the homogeneous Au–Au bowtie dimers, the electric-field enhancement exhibits a single peak at the central wavelength of 560 nm. Similar to the electric field enhancement spectrum, the plasmonic trapping potential also exhibits a two-peak structure for the heterogeneous Au–Ag bowtie dimer compared to that of Au–Au bowtie dimer with a single-peak trapping potential (Figure 4b). We can see that the highest trapping potential well can be assured for the heterogeneous Au–Ag bowtie dimer at the excitation wavelength of 415 nm. However, the robustness is poor due to the extreme sensitivity of the resonance peak to external destabilization at a wavelength of 415 nm. The robustness of the plasmonic trapping can be significantly improved for the heterogeneous Au–Ag bowtie dimer at a wavelength of 549 nm and for the homogeneous Au–Au dimer at a working wavelength of 560 nm. These results could be important for understanding the plasmonic trapping properties in the spectral regime for the noble bowtie dimers with the aim of improving the stable optical trapping ability. 

The plasmonic trapping potential tunability with respect to the edge/tip rounding of the heterogeneous Au–Ag bowtie dimer is shown in Figure 5. Here, the aspect ratio of edge/tip rounding is defined as the ratio of *L/R*, where *L* means the length of triangle edge side as seen in Figure 5b, and *R* is the radius of tip rounding. In this simulation, we fix *L* at 23.7 nm and only change the radius of tip rounding to adjust the ratio of *L/R*. We can see from Figure 5a that the dual-peak spectrum is significantly enhanced for the designed bowtie geometry with the edge/tip ratio of 11.02, in which the bowtie tip radius is taken as 2.15 nm. The smallest trapping potential at a wavelength of 549 nm reaches a maximum of 43 × 10^−21^ J, larger than 10.4 K_B_T. This indicates that the trapping potential generated at the designed bowtie dimer with the edge/tip ratio of 11.02 could be qualified for the stable trapping of nano-targets in practical applications. It can be seen that the trapping potential can be successively weakened by decreasing the edge/tip ratio from 6.77 (Figure 5b), to 4.74 (Figure 5c), to 3.65 (Figure 5d), which corresponds to tip radii of 3.5 nm, 5 nm, and 6.5 nm, respectively. However, the maximum plasmonic trapping potential of 3 × 10^−21^ J (0.7 K_B_T) would be ineffective for the edge/tip ratio of 3.65 (Figure 5d). It should be emphasized that the spectrum exhibiting definitive blue-shift with decreasing edge/tip ratio enables optimization of the trapping potential for possible applications. Additionally, the other structural control parameters could have an important influence on the plasmonic optical trapping. More results indicating the influence of structural control parameters on the plasmonic trapping are shown in Appendix A. We believe that various building geometries are very important for understanding the choice of appropriate excitation wavelengths. We also calculated the optical properties of Au nanosphere, single Au triangle, and Au–Au bowtie dimer, as shown in Appendix A. 

Figure 6 shows the calculated plasmonic trapping force on a nanosphere and corresponding potential well extracted from the center point of gap of the bowtie dimer with respect to the incident angle of the laser. The linearly polarized laser is obliquely incident at the bowtie dimer system. The incident angle is defined as the angle between the laser beam and the *y* axis of the bowtie system. We can see that the trapping force exhibits a near-linear drop with increasing laser incident angle from 0° to 90°. At an incident angle of 0°, the trapping force is 17 pN, corresponding to a trapping potential well of 72 × 10^−21^ J. However, the trapping force decreases to zero at an incident angle of 90°. Simultaneously, the plasmonic potential well exhibits a similar tendency with respect to the incident angle. In fact, the gradient electric field at the bowtie gap plays a crucial role in regulating the plasmonic optical force and potential well. Considering that the localized electric field enhancement in the bowtie gap can be sensitively dependent on polarized laser interaction with the bowtie dimer, the localized electric field would become very weak due to decoupled plasmon polarization interaction at an incident angle of 90°. As a result, the optical force and trapping potential both exhibit a decreasing tendency with increasing laser incident angle due to the degenerated electric field caused by decoupling plasmons. These results could be helpful for tuning the optical trapping properties in plasmonic bowtie dimers for the specific applications of particle-sorting, coloring, SERS imaging, and quantum lighting.

## 4. Conclusions

In summary, we theoretically investigated the symmetry-tunable plasmonic optical trapping properties of homo- and hetero-plasmonic noble metal bowtie dimer antennas. It is proposed that the trapping potential symmetry can be substantially tunable for heterogeneous Au–Ag bowtie dimers via modification of applied laser electric fields. The potential well simultaneously becomes more asymmetrical for the heterogeneous bowtie dimer at a larger laser field of 18 × 10^5^ V/m. However, the trapping potential symmetry was not influenced by the applied laser field for the homogeneous Au–Au bowtie dimer at working wavelength. These results could be helpful for advancing flexible tunable optical trapping applications in the potential fields of non-uniform nanoparticles sorting, plasmonic coloring, large-molecule imaging, etc.

## Figures and Tables

**Figure 1 nanomaterials-11-00759-f001:**
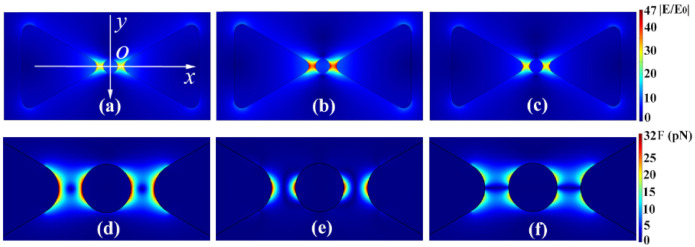
2-D images of plasmonic electric-field enhancement localized within gap of a Au–Au bowtie dimer for the different wavelengths 545 nm (**a**), 560 nm (**b**), and 600 nm (**c**), and the corresponding plasmonic optical forces presented in normalized form in (**d**). *x*-direction component (**e**), *y*-direction component (**f**). The scale bar for optical force in (**d**–**f**) is in units of pN. The bowtie edge length is 20 nm, the bowtie dimer gap is 15 nm, and a trapped 5 nm Au sphere is considered here.

**Figure 2 nanomaterials-11-00759-f002:**
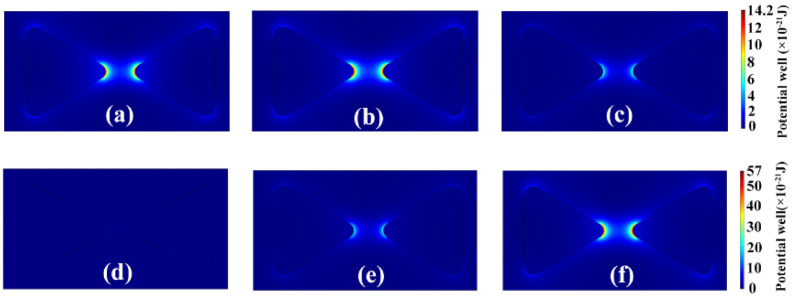
2-D images of localized plasmonic trapping potential of a homogeneous Au–Au bowtie dimer at wavelengths of 545 nm (**a**), 560 nm (**b**), and 600 nm (**c**). The laser electric field amplitudes are 2 × 10^6^ V/m, 4 × 10^6^ V/m, and 6 × 10^6^ V/m for (**d**–**f**), respectively. The scale bars have units of 10^−21^ J. The bowtie edge length is 20 nm, and the bowtie dimer gap is 10 nm.

**Figure 3 nanomaterials-11-00759-f003:**
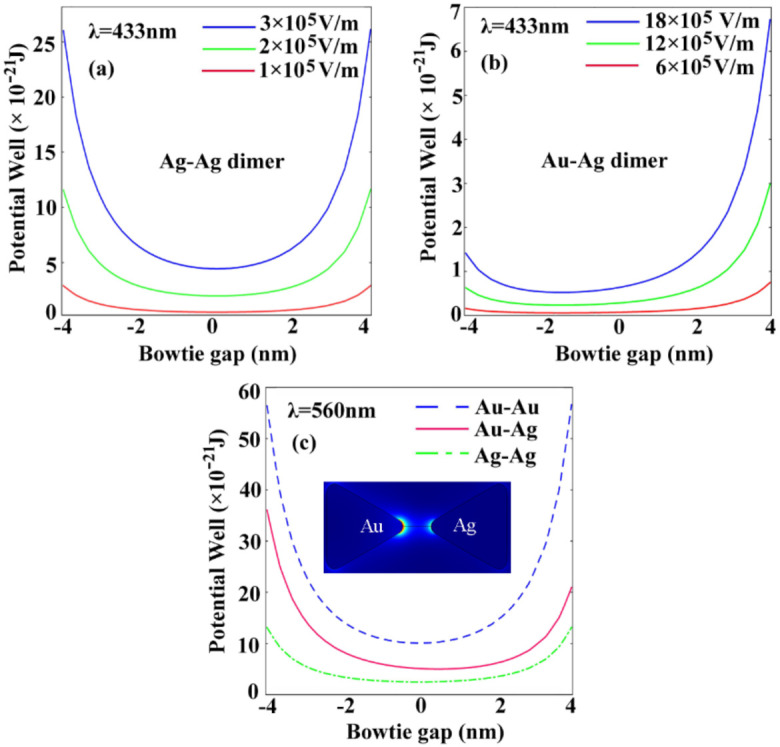
The plasmonic trapping potential distribution along the bowtie gap with respect to homogenous (**a**) and heterogeneous bowtie geometries (**b**) for different laser fields and different materials (**c**). The bowtie edge length is 20 nm, the bowtie gap is taken as 10 nm.

**Figure 4 nanomaterials-11-00759-f004:**
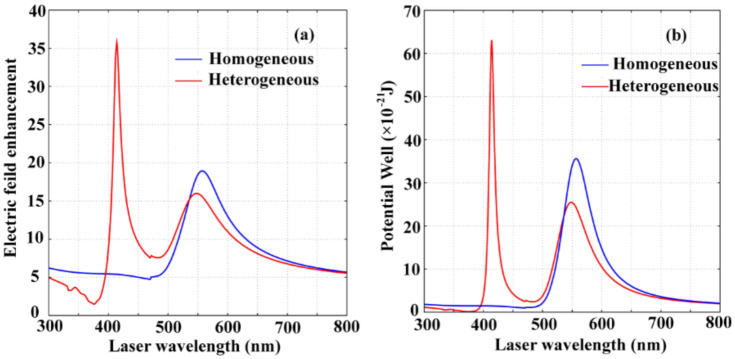
The electric-field enhancement spectra (**a**) and wavelength-dependent plasmonic potential well (**b**) for the homogeneous Au–Au and heterogeneous Au–Ag dimers. The bowtie edge length is 20 nm, the bowtie gap is taken as 10 nm.

**Figure 5 nanomaterials-11-00759-f005:**
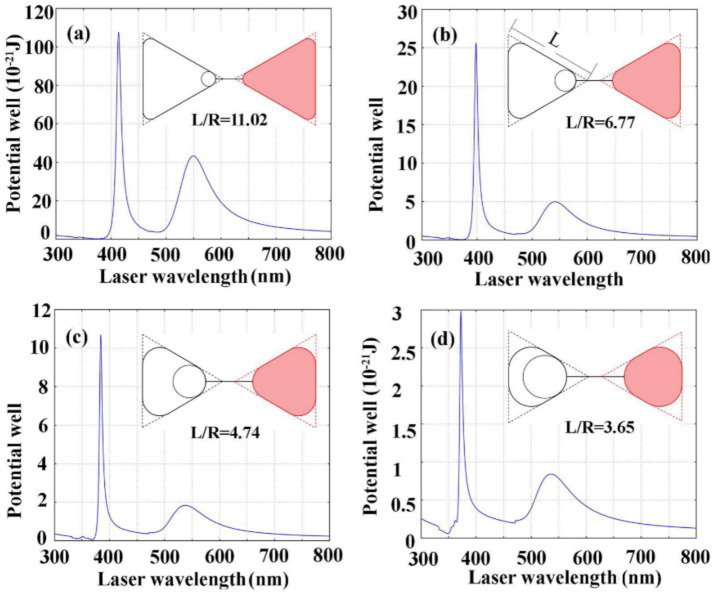
The wavelength-dependent plasmonic trapping potential tunability for the heterogeneous Au–Ag bowtie dimers with respect to the aspect ratio of edge/tip rounding. The aspect ratio of edge/tip rounding is 11.02 (**a**), 6.77 (**b**), 4.74 (**c**), and 3.65 (**d**), which corresponds to tip rounding radii of 2.15 nm, 3.5 nm, 5 nm, and 6.5 nm, respectively.

**Figure 6 nanomaterials-11-00759-f006:**
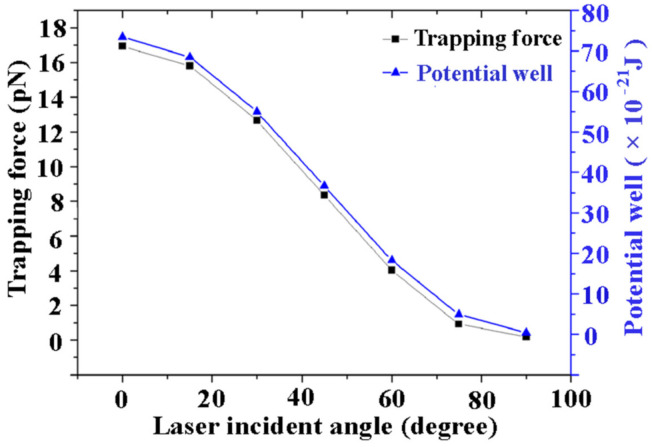
The trapping force exerted on Au nanospheres of a bowtie dimer and the corresponding potential well with respect to the incident angle of the laser. The Au sphere diameter is 10 nm here. The bowtie edge length is 40 nm, the bowtie gap is taken as 20 nm. The trapped 10 nm Au sphere is considered here.

## Data Availability

Not applicable.

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
