# Peer review of "Theoretical Study on Symmetry-Broken Plasmonic Optical Tweezers for Heterogeneous Noble-Metal-Based Nano-Bowtie Antennas"

_nanomaterials, 2021, doi:10.3390/nano11030759_

Round 1
Reviewer 1 Report
Authors Guangqing Du and coworkers report a theoretical study on the function of bowtie structures as optical traps for nanoparticles. Finite element simulations were used to investigate the potential landscape and forces resulting from plasmonic excitation. Both homo- and heterometallic bowties were considered. The study fits well with the journal Nanomaterials. Overall, I consider this paper to be interesting for a broad readership, however, after reading it, some questions remained unanswered. My recommendation is to consider these points in a revision to increase the significance of the manuscript. Tough I am strongly in favor of seeing this work published, to me, a publication in the current state seems premature.
Comments/questions:
- The title should clarify that this is a theoretical study.
- I would appreciate it if the authors could briefly mention the central structural variables (triangle dimensions, edge/tip rounding, nanosphere size, gap size) in the abstract.
- Currently, the study seems to appear a bit superficial because it refrains from discussing the influence of the main structural control parameters triangle dimensions, edge/tip rounding, nanosphere size, and gap size on the possibilities for trapping. To me, it would make perfect sense to discuss the influence of each parameter (even if only briefly) and to reason how most efficient tapping could be obtained. The authors tried to do this in respect to the gap size but also the other parameters are of importance. If the authors are hesitant to pack all of these information into the main manuscript, I suggest to provide a Supplementary Information file with more space for details.
- To facilitate the comprehension of the procedures, it would be appropriate to present the calculation of the forces in more detail, such as Eq. 1. I recommend to explain the steps in a way that scientists from other more experimental fields can easily understand them. In addition, which of the three terms has the greatest influence? Is H really the incident magnetic field or is it the induced magnetic field from the induced currents? Is it fair to assume that the contribution of the incident magnetic field is weak, in comparison to the induced electric fields? In addition, which of the three terms has the greatest influence?
- What are the structural variables applied (triangle dimensions, edge/tip rounding, nanosphere size, gap size)? I am sorry but I have a hard time to find them in the manuscript. These should be mentioned at the beginning of the Results and Discussions sections, and the reasoning behind choosing specifically these needs to be explained. Experimental studies have shown that the edge/tip rounding of triangles has a fundamental influence on the spectral properties and electromagnetic fields of triangular elements (see ACS Applied Materials & Interfaces, 10 (13), 11152–11163, 2018; and Nano Lett. 2014, 14 (8), 4810−4815). The aspect of edge/tip rounding should be discussed and reasoned in the context of single triangle and bow tie properties and trapping.
- Figures 1 and 2 would be easier to understand by addition of a scale bar for size and labeling of the two different color scales on the right. Would it maybe make sense to use kb T?
- I would like to draw the attention of the readers to few experimental studies that deal with nanoparticle decorated triangles. The readership might appreciate a stronger connection of the authors findings with the state of the art of triangle-sphere assemblies in experimental studies. This could be a nice addition to the discussions. See for instance works such as Nat Commun. 8, 14880 (2017); ACS Photonics 2020, 7, 1839; Sci Rep 6, 23203 (2016).
- The optical properties of the various building blocks (nanosphere, single triangle, and bow ties) need to be calculated and shown. These are essential for the understanding and the choice of appropriate excitation wavelengths.
Reviewer 2 Report
Du et al. reports theoretical studies on the optical properties of bowtie type Au-Au, Au-Ag, Ag-Ag dimer. Specifically, they focus on the effect of the combination of Au and Ag to the potential wells generated at the junction gap, that can be utilizable for trapping nano-objects. They claim that the heterogeneity affects the shape of the potential well, but it does not affect the trapping potential symmetry.
- This type of theoretical studies have been conducted by quite many theoretical groups. Below are what were found:
Luo et al. Optics Communications 2020, 458, 124746
Lin et al. IEEE Photonics Journal 2019, 11, 4800610
Cetin et al. International Nano Letters 2015, 5, 21.
Roxworthy et al. Nano Letters 2012, 12, 796.
Debu et al. Optical Materials Express 2020, 10, 1704.
Shoji et al. J. Phys. Chem. Lett. 2014, 5, 2957.
This topic has been an issue for last 10 years, I believe relevant and important literature should be more than what the authors cited; more discussion on the optical properties and generated force should be elaborated in Introduction.
- The experimental section is not carefully written. The authors claim that they used FEM for the analysis; since there is no cited reference, it looks like what they developed by themselves. If so, they should show how they applied the method to their analysis, and how different and how effective compared to conventional methods such as FDTD analysis and DDA analysis. If they used some commercial package, they should describe how the package is utilized. In addition, the description is not that kind to understand how they set up their systems of Au-Ag, Au-Au, and Ag-Ag. The citation is not also appropriate. It is quite tough to understand how much their method is valid with the references of 21-24. The reference 23 even does not show any equations, but it is cited for the equation in page 2.
- Some data sets are missing; Figure 1 and 2, those images for different dimers need to be provided, at least in supporting information.
- In Figure 4, what is the difference between a and b?
- Suddenly the incident angle dependence is shown. I do not get how the data is obtained from the provided experimental description. What kind of bowtie dimer is it? Is the gap of 10 nm very different from what is shown in Figure 3? It should be discussed together.
- Is the values, the magnitudes of electric field enhancement, potential well, and trapping force, comparable to the literature? More discussions with literture valeus from both theoretical and experimental studies are required to validate presented methods and results.
Round 2
Reviewer 1 Report
The authors have revised their contribution and in principle I am very satisfied with most of the changes. The clearer discussion of the structural parameters and the newly added optical data are very helpful. The new title seems most appropriate to me. I particularly welcome the new explanation of equation 1, which is done especially well, and should now be more understandable to a broader readership. However, there are some aspects that still need to be revised which I would like to briefly address them below. I recommend acceptance of the manuscript after correction of these minor issues.
- I recommend to include Figure 2 shown within the reply to comment #8 in the supporting information.
- The definition of the edge/tip rounding of triangles is not clear and probably does not coincide with the general convention. The authors state that “edge/tip rounding is defined as the ratio of L/R, where L means the length of bowtie edge side, R is the radius of tip rounding“. However, the edge length L should be the length of the unrounded triangle, at least this is the generally accepted convention (ACS Appl. Mater. Interfaces 2018, 10, 13, 11152–11163). I strongly discourage the authors to use a different definition. By defining L as the rounded edge length (as it seems to be in the current version of the manuscript), the shortening of the edge length is already included and thus the ratio L/R will follow a quite different scaling. This will make it very hard to compare the L/R values of this work with other works from past and future literature. For that reason and to avoid misunderstandings, I strongly recommend to adopt the commonly accepted definition and to recalculate the L/R values given in the manuscript.
- The schematic illustration in Figure 5a seems not be equilateral triangles. Please revise.
- The indicated edge length L seems to be incorrect (Figure 5b). Please revise in accordance with the definition given above in comment #1.
- In reference to the former comment #7, the authors have revised the text. However, the description is not clear and there could be a misunderstanding. I suggest the authors to consider that for “SERS and hot carrier devices” (which they indicated in their revised text) are not limited to bowtie geometries. In fact, individual Au triangles decorated with Au nanospheres have been proven to provide very high activities for these applications. I would encourage the authors to correct this misunderstanding and to refer to the appropriate references (e.g. by Roland Höller et al.).
Reviewer 2 Report
I am happy with the revision that the authors made. Figure 3 and Figure 4 captions: (a), (b), (c) should be described. They are missing. They may be corrected in the proof process.
